# Neuronal Migration and AUTS2 Syndrome

**DOI:** 10.3390/brainsci7050054

**Published:** 2017-05-14

**Authors:** Kei Hori, Mikio Hoshino

**Affiliations:** Department of Biochemistry and Cellular Biology, National Institute of Neuroscience, NCNP, Tokyo 187-8502, Japan; khori@ncnp.go.jp

**Keywords:** Autism, intellectual disabilities, AUTS2 syndrome, Rac1, neuritogenesis, neuronal migration, cytoskeleton, PRC1

## Abstract

Neuronal migration is one of the pivotal steps to form a functional brain, and disorganization of this process is believed to underlie the pathology of psychiatric disorders including schizophrenia, autism spectrum disorders (ASD) and epilepsy. However, it is not clear how abnormal neuronal migration causes mental dysfunction. Recently, a key gene for various psychiatric diseases, the *Autism susceptibility candidate 2* (*AUTS2*), has been shown to regulate neuronal migration, which gives new insight into understanding this question. Interestingly, the AUTS2 protein has dual functions: Cytoplasmic AUTS2 regulates actin cytoskeleton to control neuronal migration and neurite extension, while nuclear AUTS2 controls transcription of various genes as a component of the polycomb complex 1 (PRC1). In this review, we discuss AUTS2 from the viewpoint of human genetics, molecular function, brain development, and behavior in animal models, focusing on its role in neuronal migration.

## 1. The AUTS2 Gene and AUTS2 Syndrome

Accumulating studies suggest that abnormalities in cortical neuronal migration may underlie the pathology of psychiatric illnesses, including schizophrenia and autism spectrum disorders (ASDs) [1,2]. However, it is not still clear how impaired migration causes those psychiatric disorders. We believe that investigation of the *Autism susceptibility candidate 2* (*AUTS2*) gene may shed insight on this question, as this gene is related to various psychiatric diseases and also known to be involved in neuronal migration.

The *AUTS2* gene (also recently named as the “activator of transcription and developmental regulator” by the HUGO Gene Nomenclature Committee (HGNC), #14262) located on chromosome 7q11.22, was initially linked with ASDs in a study by Sultana et al., who reported that the *AUTS2* gene locus was disrupted by a *de novo* balanced translocation in two monozygotic twins with ASDs [3]. Extensive clinical genomic analyses have subsequently identified more than 50 additional patients with *AUTS2* mutations, who presented similarly, but with high variability in the severity and combination of phenotypes of not only the autistic features, but also borderline to mild intellectual disabilities (IDs), microcephaly, feeding difficulties, short stature, hypotonia, and cerebral palsy as well as the dysmorphisms in craniofacial features [4,5,6]. These common pathological features including ID, microcephaly and craniofacial abnormalities, except for the autistic features among the individuals with *AUTS2* mutations, have been recently termed “AUTS2 syndrome” [5,7]. Moreover, the genomic structural variants and single nucleotide polymorphisms (SNPs) in the *AUTS2* locus are also associated with a wide range of other neurological disorders such as epilepsy, schizophrenia, attention deficit hyperactivity disorder (ADHD), dyslexia and depression as well as addiction-related behaviors including drug addiction and alcohol consumption, implicating that *AUTS2* is broadly involved in neurodevelopment [8,9,10,11,12,13,14,15,16].

*AUTS2* is one of the largest genes in mammals, spanning 1.2 Mb and containing 19 exons (Figure 1A) [17]. The first six exons at the 5′ end are separated with large introns, whereas the remaining 13 exons, which are highly conserved in vertebrates, are compact with smaller clustered introns at the 3′ end. The main transcript of *Auts2* encodes a relatively large protein (1259 aa for human (NM_015570) and 1261 aa for mus musculus (NM_177047)), but multiple alternatively spliced isoforms also exist. The representative five distinct isoforms of mouse AUTS2 are shown in Figure 1B. Beunders and colleagues have identified a short 3′ *AUTS2* human mRNA transcript that originates from exon 9 [5]. Immunoblot experiments and 5′ RACE (5′ rapid amplification of cDNA ends) analysis revealed the existence of C-terminal AUTS2 short isoform variants that are translated from translational start sites in the middle of exons 8 and 9 in the mouse developing brain [18]. In addition to the C-terminal AUTS2 short isoforms, two short isoforms containing the first three (uc008zuy.1) or five (uc008zuw.1) exons at the 5′ end encoding the N-terminus of mouse AUTS2 are also annotated in the UCSC Genome bioinformatics (the University of California Santa Cruz, https://genome.ucsc.edu), although the expression of these proteins remain to be confirmed. Western blot analysis showed that the full-length AUTS2 isoform begins to be expressed from the early neurodevelopmental stage at embryonic day 12 (E12) in the mouse brain and is continuously present postnatally, although the expression level drops after birth [18,19]. In contrast, the C-terminal AUTS2 short isoforms transiently appear prenatally with peak expression at mid-neurodevelopment (around E14) and then disappear postnatally [18]. Therefore, the expression of AUTS2 isoforms is differentially regulated during the brain development.

In the developing mouse brain, *Auts2* is broadly expressed in multiple regions, but particularly high expression is found in the regions associated with higher cognitive brain functions, including the neocortex, Ammon’s horn and dentate gyrus of hippocampus as well as the cerebellum in prenatal brains [20]. In mature brains, the expression of *Auts2* is restricted to a few types of neural cells including the pyramidal CA neurons in the hippocampus, granule neurons of the dentate gyrus as well as the cerebellar Purkinje cells. Interestingly, the expression of *Auts2* is confined to the prefrontal region in the cerebral cortex [20,21]. *Auts2* expression is reportedly regulated by a transcription factor, Tbr1 [21].

## 2. AUTS2 Functions in Cytoskeletal Regulation and Transcriptional Activation

The biological function of AUTS2 in brain development has long remained unclear, but two groups initially performed the molecular studies for AUTS2 using zebrafish models. Knockdown of zebrafish *auts2* by morpholino led to various abnormalities including microcephaly, reduction of neural cells and movement disorders as well as craniofacial dysmorphisms [5,22]. Furthermore, the full-length or the 3′ end short transcripts of human *AUTS2* were found to rescue several abnormalities including the microcephaly and dysmorphisms in the morphant fish, suggesting that the C-terminal region of AUTS2 contains the crucial elements for neurodevelopment [5]. In humans, it has been reported that the severity of the pathological features of the AUTS2 syndrome is higher in individuals with the deletions of exons at the 3′ of the *AUTS2* locus compared to the different combinations of in-frame deletions of exons 2–5 at the 5′ region [5]. Some exceptions, however, have also been reported, such as that individuals with a deletion of exon 1 or a frame shift deletion at exon 6 displayed a high severity score of AUTS2 syndrome [23,24].

In the mouse brain, AUTS2 is exclusively present in the cell nuclei during early developmental stages, but AUTS2 also appears in cytoplasm as well as dendrites and axons of the differentiated neurons as neurodevelopment proceeds [18]. The subcellular localization analyses for the recombinant AUTS2 proteins exogenously expressed in the neuroblastoma cells revealed that the full length AUTS2 or its N-terminal truncated protein fragments containing the proline-rich domain (PR1) are localized in both cell nuclei and cytoplasm including the growth cones and neurites as well as the cell bodies in differentiated neural cell lines. In contrast, the C-terminal fragments including the corresponding AUTS2 short isoform are exclusively nuclear [18].

Recent studies conducted by two groups provided insight into the function of nuclear AUTS2 as a transcriptional regulator for neural development. Oksenberg et al. performed analyses using the combination of chromatin immunoprecipitation followed by high throughput sequencing (ChIP-seq) and transcriptome analysis by RNA-sequencing to investigate the role of AUTS2 in gene regulatory networks in mouse embryonic forebrain [25]. They found that AUTS2 interacts with the promoters/enhancers of various genes that are related to brain development and also associated with neurological disorders. Furthermore, the studies by Gao and colleagues showed that AUTS2 acts as a co-factor of Polycomb Repressive Complex 1 (PRC1) and activates gene expression [19]. Interestingly, it has been demonstrated that PRC1 canonically acts as a transcriptional repressor, but the participation of AUTS2 in PRC1 turns this complex into a transcriptional activator by recruiting histone acetyltransferase p300, a well-known transcriptional co-activator, as well as casein kinase 2, which inhibits the repressive function of PRC1 [19].

In the cytoplasm, AUTS2 interacts with guanine nucleotide exchange factors (GEFs), P-Rex1 and Elmo2/Dock180 complex, to activate a Rho family small GTPase, Rac1, a key coordinator of actin polymerization and microtubule dynamics [18,26,27]. The cytoplasmic AUTS2 promotes lamellipodia formation in neuroblastoma cells and neurite extension of hippocampal primary cultured neurons in vitro. On the other hand, AUTS2 interacts with other GEFs, intersectin (ITSN) 1 and ITSN2, to suppress the activities of another Rho family GTPase, Cdc42, leading to repression of filopodia formation in the neurites and cell bodies of neurons [18,28].

## 3. AUTS2 Functions in Neuronal Migration and Neurite Extension

In the developing cerebral cortex in mice, AUTS2 is expressed in radially migrating neurons in the intermediate zone (IZ) and the cortical plate (CP) as well as maturating neurons in the CP, but barely in progenitors in the ventricular zone [18,20]. In those neurons, AUTS2 protein localizes not only in cell nuclei but also in cytoplasmic regions, including their processes. The knockdown and/or knockout of *Auts2* in neurons in embryonic mouse brains results in impairment and retardation of their migration [18]. However, this abnormality is rescued by co-introduction of the full-length AUTS2 but not by short isoform AUTS2 [18]. This indicates that only the full-length isoform is involved in neuronal migration, although both isoforms are expressed in migrating neurons. Furthermore, NES (nuclear export sequence)-tagged AUTS2 that exclusively localizes at extra-nucleic regions of cells is able to rescue the impairment [18]. This observation further implies that AUTS2 functions in the cytoplasm and/or processes functions for neuronal migration, although the AUTS2 protein resides both in nucleic and extra-nucleic regions of migrating neurons.

As mentioned in the previous section, AUTS2 can activate Rac1 and inactivate Cdc42 [18]. It was suggested that the AUTS2-Rac1 pathway is involved in cortical neuronal migration because the introduction of wild type Rac1 into *Auts2* KO neurons was able to rescue their abnormal migration [18]. On the other hand, as a dominant negative form for Cdc42 could not rescue the abnormality [18], it was implied that negative regulation of Cdc42 by AUTS2 might not play a large role in cortical neuronal migration. It was previously shown that a few Rac GEFs, such as P-Rex1, STEF (Tiam2) and Tiam1, participate in cortical neuronal migration through activating Rac1 [26,29]. Because P-Rex1 but not STEF or Tiam1 are involved in the AUTS2-Rac1 pathway [18], it is deduced that P-Rex1 relays the signal from AUTS2 to Rac1 for neuronal migration.

Migrating *Auts2* mutant neurons displayed abnormal morphologies in their processes during the transition from a multipolar to a bipolar state at the upper-intermediate zone [18]. In addition, the activity of JNK was severely reduced in those neurons. Among JNK isoforms, JNK2 was reported to regulate cortical neuronal migration [30]. It was previously shown that, under the control of Rac1, JNK is involved in leading the process formation of migrating neurons through the regulation of microtubules dynamics [29]. On the other hand, it was reported that the morphology of multipolar and leading processes of migrating neurons are regulated by actin cytoskeletal reorganization, which is mediated by actin binding proteins Cofilin and Filamin A. Cofilin and Filamin A are downstream and binding partners of Rac1, respectively [31,32,33]. Taken together, this suggests that the AUTS2-Rac1 pathway dynamically regulates the cytoskeleton, including actin and microtubules, for proper neuronal migration, as schematically depicted in Figure 2.

The introduction of AUTS2 into primary cortical neurons induced neurite elongation [18]. However, this effect was blocked by the co-introduction of dominant negative forms of Rac1 or Elmo2, but not that of P-Rex1. Furthermore, in vivo, axon elongation of cortical commissural neurons, as visualized by electroporated enhanced green fluorescent protein (EGFP), was impaired in *Auts2* KO brains [18]. This could be rescued by wild type Rac1. These findings suggest that AUTS2-Elmo2/Doc180 is involved in axonogenesis/neuritogenesis during cortical development.

In heterozygous mutants for AUTS2, mild abnormalities were observed in cortical neuronal migration and neurite extension. As most psychiatric patients with *AUTS2* mutations carry heterozygous mutations, similar abnormalities in neuronal migration and/or neurite extension may occur in those patients. Of note, some patients with *AUTS2* mutations exhibit corpus callosum hypoplasia, in addition to psychiatric phenotypes [34].

## 4. Behaviors of AUTS2 Mutant Mice

In mice, homozygous pups with neuron-specific ablation of the full-length *Auts2* isoform display abnormalities in motor skills including defects in righting reflex and negative geotaxis as well as reduction of ultrasonic vocalization following maternal separation [19]. Furthermore, adult heterozygotes of *Auts2* global knockout exhibit neurocognitive defects in several behavioral tests [35]. The *Auts2* heterozygous mutant mice show reduced exploratory activity in a novel circumstance as well as lower anxiety against aversive circumstances such as an elevated platform or illuminated open area. Furthermore, recognition and associative memory were impaired in the *Auts2*-heterozygous mice, suggesting that AUTS2 plays critical roles in emotional control as well as learning and memory formation. These behavioral phenotypes mimic some of the symptoms of human patients with heterozygous *AUTS2* mutations, indicating that these mutant mice may be good animal models to study the pathology of psychiatric illnesses.

## 5. Conclusions and Future Perspective

The acquisition of cognitive brain functions is achieved by the proper progression of a series of neurodevelopmental events that are controlled by a complex network of genetic programs. Neuronal migration is one of the most important initial steps during the organization of the cerebral cortex architecture, and a large number of intrinsic and extrinsic factors are intricately involved in the regulation of this event. Since its identification in 2002, *AUTS2* has emerged as a crucial gene associated with a wide range of neurodevelopmental and psychological disorders including ASDs, ID and schizophrenia as well as drug addiction and alcohol consumption. The physiological roles of AUTS2 in neural development have not, however, been clarified despite the fact that multiple studies have implicated the involvement of AUTS2 in brain development.

Over the past few years, molecular analyses and behavioral studies using model organisms such as zebrafish and mice have helped to elucidate the molecular functions of AUTS2 with regard to gene expression and the regulation of cytoskeletal dynamics during neural development. The nuclear AUTS2 acts as a transcriptional activator for neuronal gene expression by interacting with the epigenetic modulator, PRC1. In extra-nucleic regions of the developing cortical neurons, AUTS2 plays a critical role in the regulation of Rac1 signaling in several neurodevelopmental processes, including cortical neuronal migration and subsequent neurite formation. These studies together with previous works have provided insight into the potential roles of this gene in the acquisition of neurocognitive brain functions. Moreover, given that AUTS2 is continuously expressed during postnatal and mature brains, AUTS2 may be involved in other neurodevelopmental processes such as synapse formation, assembly of neural circuits and the functionally distinct cortical area formation. Further studies will be required to elucidate the functions of AUTS2 in postnatal brain development.

Many other questions also remain, however, regarding the pathogenic mechanisms underlying the occurrence of various types of neuropsychiatric illnesses in individuals with *AUTS2* mutations as well as the phenotypic variations in AUTS2 syndrome. In addition, although several lines of evidence have demonstrated that the expression and the cellular localization of AUTS2 isoforms may be tightly controlled during the neurodevelopment in a spatiotemporally regulated manner, the regulatory mechanisms of expression, the physiological roles in neural development, and the involvement of psychiatric disorders with the distinctive AUTS2 isoforms remains to be elucidated. Future studies decoding these enigmas for AUTS2 will not only provide us with fundamental knowledge about neurodevelopment, but also help to delve into the pathological mechanisms underlying neuropsychiatric illnesses by AUTS2 dysregulation.

## Figures and Tables

**Figure 1 brainsci-07-00054-f001:**
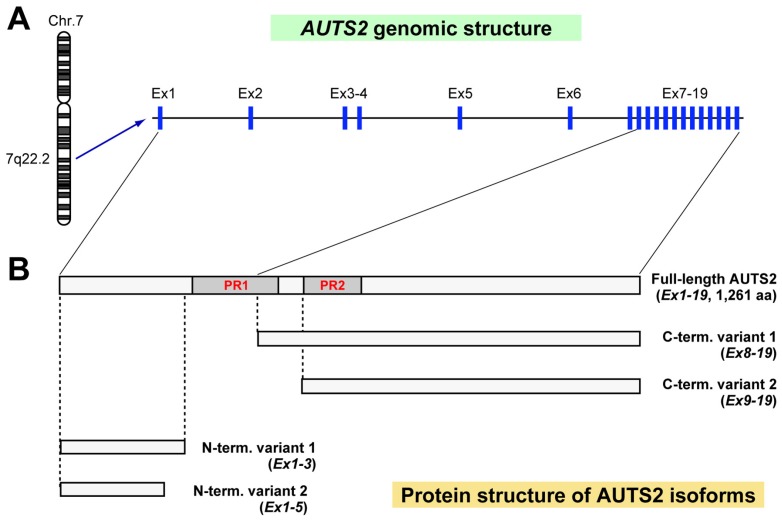
Schematic of *AUTS2* genomic region and the protein structure of AUTS2 isoforms. Genomic structure of human *AUTS2* locus in chromosome 7 (Chr.7) (**A**) and protein structure of the representative five different mouse AUTS2 isoforms (**B**) are depicted. The exons (blue squares in A) corresponding to the protein regions are indicated with solid lines. Ex: exon, PR: proline-rich domain.

**Figure 2 brainsci-07-00054-f002:**
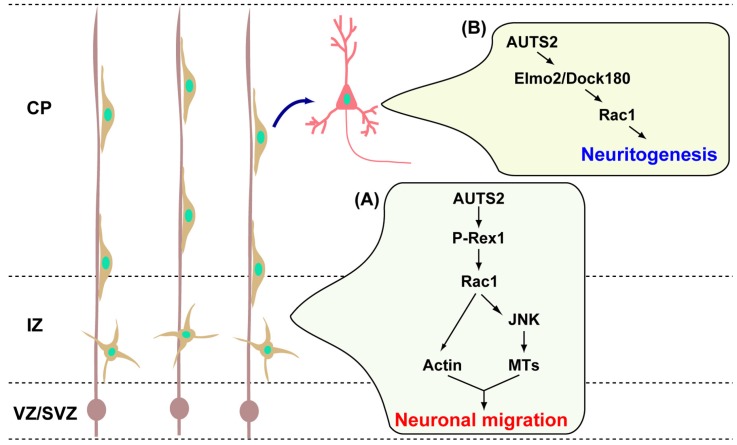
AUTS2-Rac signaling pathways in neuronal migration and neuritogenesis. (**A**) At the developing cerebral cortex, AUTS2 acts as an upstream factor for the P-Rex1-Rac1 signaling pathway to control the neuronal migration regulating the cytoskeletal rearrangements; (**B**) After completion of migration, AUTS2 activates Rac1 via the Elmo2/Dock180 complex and promotes the neurite extensions of cortical neurons. CP: cortical plate, IZ: intermediate zone, JNK: c-Jun N-terminal kinase, MTs: microtubules, VZ/SVZ: ventricular/subventricular zone.

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
