# Peer review of "Neuronal Migration and AUTS2 Syndrome"

_brainsci, 2017, doi:10.3390/brainsci7050054_

Round 1

Reviewer 1 Report

This is a well written short review on one of the most interesting genes present in humans and other species. The appearance of this review is timely. Most of the relevant literature has been included.

My major concern is that an overall conclusion at the end of the review is missing. The authors are specialists in the field, and they know their subject well. Therefore, it would be interesting if they would elaborate a little more on the combined meaning of all the individual facts they describe. This paper would certainly benefit from a little bit of scientifically funded speculation.

other remarks:

-line 14, repeated further in the manuscript. You mention psychiatric illnesses. I'm not sure if this is the correct term to use, maybe neurological illnesses would be better.

-line 32, please check is AUTS2 gene, when combined with word gene, should be in italics.

-line 38, it is important to mention that classic autism is not a part of AUTS2 syndrome

-line 59, the part about the isoforms is puzzling. It would be helpful if you add the identifiers of the transcripts you refer to. If you look in UCSC for AUTS2 many more isoforms can be found. Furthermore, you mention that only bioinformatic evidence is present for the N-terminal isoforms, whereas in line 93 you mention immunohistochemistry studies concerning the N-terminal isoforms. This part of the text should be improved and clarified.

-legend to figures 1 and 2. These are much too short. It should be explained what is depicted in the figure, without the need to go back to the text of the manuscript. Furthermore, it should be indicated what the abbreviations mean.

-line 85. The milder phenotype is associated with in frame deletions of exons 2-5 (in different combinations). Exon 1 deletions, as well as the out of frame exon 6 deletions, have a severe phenotype, so it's not C-terminal deletions in general that are milder.

-line 100-101, sentence is difficult to read, please rephrase

-line 107, "as well as"  should be "as well as in"

-line 122-131, this is just a list of facts,  it's unclear from which reference they are coming, and also why they are mentioned here (no context). This part should be improved

-Line 134. JNK should be introduced. Furthermore, there are many JNK isoforms and genes, which ones are you writing about? please give more detail and add a more recent reference

-Figure 2, it looks like P-Rex1 has an essential role, but this is not mirrored in the body of the text. It should be explained better why this protein has such an important position in this figure

Author Response

Response to Comments by Reviewer#1

    We would like to thank the Reviewers for taking time to read our manuscript carefully.

We believe that we have adequately addressed the issues brought up by the Reviewer#1.

Point-by-point discussions of the changes re listed below. The Reviewer's comments are indicated in bold.

(1) My major concern is that an overall conclusion at the end of the review is missing. The authors are specialists in the field, and they know their subject well. Therefore, it would be interesting if they would elaborate a little more on the combined meaning of all the individual facts they describe. This paper would certainly benefit from a little bit of scientifically funded speculation.

According to the reviewer’s suggestion, we changed the subheading of Section 5 to “Conclusions and Future Perspective” and added several paragraphs (below and in Lines 194-215) as conclusions:

Acquisition of the cognitive brain functions is achieved by proper progression of a series of neurodevelopmental events that are controlled by a complex network of genetic programs. Neuronal migration is one of the most important initial steps during the organization of the cerebral cortex architecture, and a large number of intrinsic and extrinsic factors are intricately involved in the regulation of this event. Since its identification in 2002, AUTS2 has emerged as a crucial gene associated with a wide range of neurodevelopmental and psychological disorders including ASDs, ID and schizophrenia as well as drug addiction and alcohol consumption. The physiological roles for AUTS2 in neural development have not long been, however, clarified despite the fact that multiple studies have implicated the involvement of AUTS2 in CNS development.

 Over the past few years, molecular analyses and behavioral studies using model organisms such as zebrafish and mice have helped to elucidate the molecular functions of AUTS2 with regard to gene expression and the regulation of cytoskeletal dynamics during neural development. The nuclear AUTS2 acts as a transcriptional activator for neuronal gene expression by interacting with the epigenetic modulator, PRC1. In extranucleic regions of the developing cortical neurons, AUTS2 plays a critical role in the regulation of Rac1 signaling in several neurodevelopmental processes including cortical neuronal migration and subsequent neurite formation. These studies together with previous works have provided insight into the potential roles of this gene in the acquisition of neurocognitive brain functions. Moreover, given that AUTS2 is continuously expressed during postnatal and mature brains, AUTS2 may be involved in other neurodevelopmental processes such as synapse formation, assembly of neural circuits and the functionally distinct cortical area formation. Further studies will be required to elucidate the functions of AUTS2 in postnatal brain development.”

(2) Line 14, repeated further in the manuscript. You mention psychiatric illnesses. I'm not sure if this is the correct term to use, maybe neurological illnesses would be better.

The word "neurological illnesses" includes neurodegenerative disorders to which the AUTS2 gene is not related. Therefore, we changed the phrase "psychiatric illnesses" to "psychiatric diseases" in Abstract section.

(3) Line 32, please check is AUTS2 gene, when combined with word gene, should be in italics.

We agree and have added Italics.

(4) Line 38, it is important to mention that classic autism is not a part of AUTS2 syndrome.

We added the phrase “except for the autistic features” in the sentence describing the AUTS2 syndrome in line 41-42.

(5) Line 59, the part about the isoforms is puzzling. It would be helpful if you add the identifiers of the transcripts you refer to. If you look in UCSC for AUTS2 many more isoforms can be found. Furthermore, you mention that only bioinformatic evidence is present for the N-terminal isoforms, whereas in line 93 you mention immunohistochemistry studies concerning the N-terminal isoforms. This part of the text should be improved and clarified.

According to the suggestion, we have added the following accession numbers for the transcripts of AUTS2/Auts2 gene:

Human full-length AUTS2 [NM_0155701]

Mouse full-length Auts2 [NM_177047]

N-terminal Auts2 short isoform composed with 3 exons [uc008zuy.1] (UCSC)

N-terminal Auts2 short isoform composed with 5 exons [uc008zuw.1] (UCSC)

We also changed the phrase in lines 51-53 to reflect the reviewer’s comment that more transcripts are annotated than the ones we described:

“The main transcript of Auts2 encodes a relatively large protein (1,259 aa for human (NM_015570] and 1,261 aa for mus musculus (NM_177047)), but multiple alternatively spliced isoforms also exist. The representative five distinct isoforms of mouse AUTS2 are shown in Figure 1B.”

With regard to the immunohistochemical studies for the subcellular localization of AUTS2 protein, we apologize that the description in our manuscript was unclear. In this experiment, the recombinant full-length AUTS2 proteins as well as the various types of truncated mutant proteins including the N-terminal portion of AUTS2 (different from the N-terminal AUTS2 short isoform) were exogenously expressed in a neuroblastoma cell line (N1E-115 cells). Therefore, we modified the phrase in lines 99-103 as follows:

“The subcellular localization analyses for the recombinant AUTS2 proteins exogenously expressed in the neuroblastoma cells revealed that the full length AUTS2 or its N-terminal truncated protein fragments containing the proline-rich domain (PR1) are localized in both cell nuclei and cytoplasm including the growth cones and neurites as well as the cell bodies in differentiated neural cell lines.”

(6) Legend to figures 1 and 2. These are much too short. It should be explained what is depicted in the figure, without the need to go back to the text of the manuscript. Furthermore, it should be indicated what the abbreviations mean.

According to the referee’s suggestion, we have added more detailed explanations for the models in Figure legends 1 and 2 in the manuscript.

(7) Line 85. The milder phenotype is associated with in frame deletions of exons 2-5 (in different combinations). Exon 1 deletions, as well as the out of frame exon 6 deletions, have a severe phenotype, so it's not C-terminal deletions in general that are milder.

We have now modified the explanation about AUTS2 deletions from the protein regions (N and C-terminus) to the genomic locus (5’ and 3’). We also added more information about the types of mutations (in-frame and out-frame mutations) as well as the exceptional cases for AUTS2 mutations in the sentence suggested by the reviewer as follows (Line 91-96):

“In humans, it has been reported that the severity of the pathological features of the AUTS2 syndrome is greater in individuals with the deletions of exons at the 3’ of the AUTS2 locus compared to the different combinations of in-frame deletions of exons 2-5 at the 5’ region. Some exceptions have been, however, also reported that individuals with a deletion of exon 1 or a frame shift deletion at exon 6 displayed a high severity score of AUTS2 syndrome.”

(8) Line 100-101, sentence is difficult to read, please rephrase.

As suggested, the sentence was changed to the following:

“They found that AUTS2 interacts with the promoters/enhancers of various genes that are related to brain development and associated with neuropsychiatric disorders.”

(9) Line 107, "as well as" should be "as well as in"

We have corrected the phrase.

(10) Line 122-131, this is just a list of facts, it's unclear from which reference they are coming, and also why they are mentioned here (no context). This part should be improved.

We revised the sentences indicated by the reviewer as described below as well as in the manuscript (Line 131-146).

“However, this abnormality is rescued by co-introduction of the full-length AUTS2 but not by short isoform AUTS2 [18]. This indicates that only the full-length isoform is involved in neuronal migration, although both isoforms are expressed in migrating neurons. Furthermore, NES (nuclear export sequence)-tagged AUTS2 that exclusively localizes at extranucleic regions of cells is able to rescue the impairment [18]. This observation further implies that AUTS2 functions in the cytoplasm and/or processes for neuronal migration, although the AUTS2 protein resides both in nucleic and extranucleic regions of migrating neurons.

  As mentioned in the former section, AUTS2 can activate Rac1 and inactivate Cdc42 [18]. It was suggested that the AUTS2-Rac1 pathway is involved in cortical neuronal migration because introduction of wild type Rac1 into Auts2 KO neurons was able to rescue their abnormal migration [18]. On the other hand, as a dominant negative form for Cdc42 could not rescue the abnormality [18], it was implied that negative regulation of Cdc42 by AUTS2 may not play a large role in cortical neuronal migration.

 It was previously shown that a few Rac GEFs, such as P-Rex1, STEF (Tiam2) and Tiam1, participate in cortical neuronal migration through activating Rac1 [24, 27]. Because P-Rex1 but not STEF or Tiam1 are involved in the AUTS2-Rac1 pathway [18], it is deduced that P-Rex1 relays the signal from AUTS2 to Rac1 for neuronal migration.”

(11) Line 134. JNK should be introduced. Furthermore, there are many JNK isoforms and genes, which ones are you writing about? Please give more detail and add a more recent reference

Although there have been several studies about JNK and migration, the situation is a bit complicated: Some are contradictory to others. Therefore we simplified the description of JNK in the text. We added a sentence about the JNK isoform at line 149-150 in the manuscript as follows:

"Among JNK isoforms, JNK2 was reported to regulate cortical neuronal migration.”

(12) Figure 2, it looks like P-Rex1 has an essential role, but this is not mirrored in the body of the text. It should be explained better why this protein has such an important position in this figure

Together with the question (9), we added the following explanation for it at line 143-146 in the manuscript:

"It was previously shown that a few Rac GEFs, such as P-Rex1, STEF (Tiam2) and Tiam1, participate in cortical neuronal migration through activating Rac1 [24, 27].  Because P-Rex1 but not STEF or Tiam1 are involved in AUTS2-Rac1 pathway [18], it is deduced that P-Rex1 relays the signal from AUTS2 to Rac1 for neuronal migration."

Reviewer 2 Report

In this review, the authors focused on the role of AUTS2 in neuronal migration. They clearly summarized the current knowledge about AUTS2 in the aspect of brain development, neuronal migration and AUTS2 syndrome. The abstract is well written and clearly states what the review is about. The review is short, simple, well structured, and clearly states the findings and facts about AUTS2 in mammalian brain. 

Author Response

Response to Comments by Reviewer#2

   We would like to thank the Reviewers for taking time thread our manuscript carefully.

We revised our manuscript according to the comments by Reviewer#1.

We believed that we have adequately addressed the issues brought up by the reviewer and fell that the revisions have further improved our paper.
